# Dynamics-inspired Structure Hallucination for Protein-protein Interaction Modeling

## Abstract

Protein-protein interaction (PPI) represents a central challenge within the biology field, and accurately predicting the consequences of mutations in this context is crucial for drug design and protein engineering. Deep learning (DL) has shown promise in forecasting the effects of such mutations but is hindered by two primary constraints. First, the structures of mutant proteins are often elusive to acquire. Secondly, PPI takes place dynamically, which is rarely integrated into the DL architecture design. To address these obstacles, we present a novel framework named Refine-PPI with two key enhancements. First, we introduce a structure refinement module trained by a mask mutation modeling (MMM) task on available wild-type structures, which is then transferred to hallucinate the inaccessible mutant structures. Second, we employ a new kind of geometric network, called the probability density cloud network (PDC-Net), to capture 3D dynamic variations and encode the atomic uncertainty associated with PPI. Comprehensive experiments on SKEMPI.v2 substantiate the superiority of Refine-PPI over all existing tools for predicting free energy change. These findings underscore the effectiveness of our hallucination strategy and the PDC module in addressing the absence of mutant protein structure and modeling geometric uncertainty.

## 1 Introduction

Proteins seldom act in isolation and typically engage in interactions with others to perform a wide array of biological functions (Phizicky & Fields, 1995; Du et al., 2016). One illustrative instance involves antibodies, which belong to a protein category within the immune system. They identify and attach to proteins found on pathogen surfaces and trigger immune responses by interacting with receptor proteins in immune cells (Lu et al., 2018). Accordingly, it is crucial to devise approaches to modulate these interactions, and a prevalent strategy is to introduce amino acid mutations at the interface (see Fig. 1). However, the space of possible mutations is vast, making it impractical or prohibitive to conduct experimental tests on all viable modifications in a laboratory setting (Li et al., 2023). Thus, computational techniques are required to guide the recognition of desirable mutations by forecasting their mutational effects on binding strength, commonly measured by the change in binding free energy termed $\Delta\Delta G$.

The past decade has witnessed the great potential of deep learning (DL) techniques (Rives et al., 2021; Min et al., 2022) in biological science, such as protein design (Jing et al., 2020), folding classification (Hermosilla et al., 2020), model quality assessment (Wu et al., 2023), and function prediction (Gligorijević et al., 2021). These DL algorithms also surpass conventional approaches in computing $\Delta\Delta G$ and can be roughly divided into biophysics- and statistics-based kinds. In particular, the former depends on sampling from energy functions and consequently faces a trade-off between efficiency and accuracy (Schymkowitz et al., 2005; Leman et al., 2020). Meanwhile, statistical-based methods are limited by the selection of descriptors and cannot take advantage of the growing availability of protein structures (Alford et al., 2017).

Despite DL's fruitful progress in identifying $\Delta\Delta G$, their efficacy encounters various obstacles. First is the absence of the mutant complex structure. Due to the long-standing consensus that protein function is intricately related to its structure (Jumper et al., 2021), an emerging line seeks to encode protein structures using 3D-CNNs or GNNs (Jing et al., 2020; Satorras et al., 2021), but typically relies on experimental structures like Protein Data Bank (PDB). Their performance deteriorates

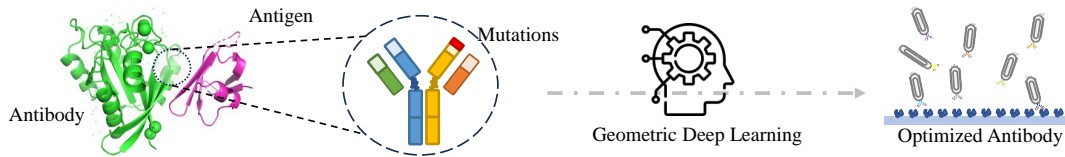

Figure 1: Geometric deep learning is applied to optimize the antibody sequences and achieve desired properties (*e.g.*, better affinity and specificity).

significantly when fed low-quality or noisy protein structures (Huang et al., 2024). Regrettably, in antibody optimization, obtaining mutant structures is an insurmountable obstacle, and the exact conformational variations upon mutations are unknown. While groundbreaking approaches such as Alphafold (Jumper et al., 2021) and Alphafold-Multimer (Evans et al., 2021) have brought a revolution in directly inferring protein structures from amino acid sequences, they struggle to accurately forecast the structure of antibody-antigen complexes compared to monomers (Ruffolo et al., 2023). As an alternative, some scientists turn to energy-based protein folding tools like FoldX (Delgado et al., 2019) to sample mutant structures, which show finite efficacy and dramatically increase overall computational time (Cai et al., 2023). The second limitation is the overlook of existing DL on the fundamental thermodynamic principle. Proteins exhibit inherent dynamism, critical for biological functions and therapeutic targeting (Miller & Phillips, 2021). Many real-world observations are not solely dependent on a single structure but influenced by the equilibrium distribution (Ganser et al., 2019). For example, inferring biomolecule functions involves assessing the probabilities associated with various structures to identify metastable states.

To overcome these barriers, we introduce Refine-PPI (see Fig. 3) with two key innovations for the mutation effect prediction problem. First, we devise a masked mutation modeling (MMM) strategy and propose to predict the mutant structure and $\Delta\Delta G$ simultaneously. Refine-PPI combines the prediction of structure and the prediction of free energy change into a joint training objective rather than relying on external software to sample mutant structures. This offers several distinct advantages. On the one hand, the hallucinated mutant structure exhibits significant differences from the wild-type structure, providing crucial geometric information related to the change in binding free energy. On the other hand, MMM not only enables inference of the most likely equilibrium conformation of the mutant structure but also encourages graph manifold learning with the denoising objective Godwin

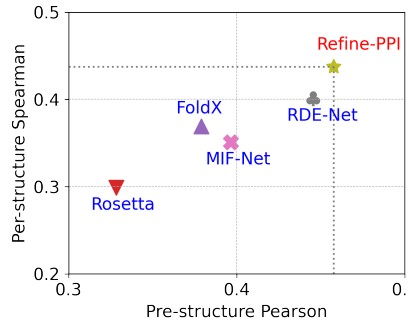

Figure 2: Performance of Refine-PPI on SKEMPI.v2 compared to other energy-based or pretrained baselines.

et al. (2021). Besides, $\Delta\Delta G$ implicitly conveys extra information about the structural difference before and after the mutation. Collective training with $\Delta\Delta G$ would promote the efficiency of structure prediction. Second, we introduce a new kind of geometric GNN called PDC-Net to capture the flexibility and dynamics of conformations during the binding process. Specifically, each particle in a complex is represented as a probability density cloud (PDC) that illustrates the scale and strength of their motion throughout the interaction procedure. Then, an aligned network is used to propagate the distributions of the equilibrium of molecular systems. A comprehensive evaluation in the SKEMPI.v2 dataset (Jankauskaitė et al., 2019) proves that our Refine-PPI outperforms all present methodologies by a significant margin (see Fig. 1) and it is promising to generate absent mutant structures via a multi-task training scheme.

## 2 PRELIMINARY AND BACKGROUND

**Definition and Notations.** A protein-protein complex is a multi-chain protein structure, separated into two groups. Each group contains at least one protein chain and each chain consists of several amino acids. The wild-type complex is represented as a 3D graph $\mathcal{G}^{\mathrm{WT}}$, constituted of a ligand $\mathcal{G}_{\mathrm{L}}^{\mathrm{WT}}$ and a receptor $\mathcal{G}_{\mathrm{R}}^{\mathrm{WT}}$. $\mathcal{G}$ is composed of a batch of nodes $\mathcal{V}$ and edges $\mathcal{E}$. $\mathcal{V}$ represents residues or atoms at different resolutions, and $v_i \in \mathcal{V}$ has several intrinsic attributes such as the initial $\psi_h$-dimension

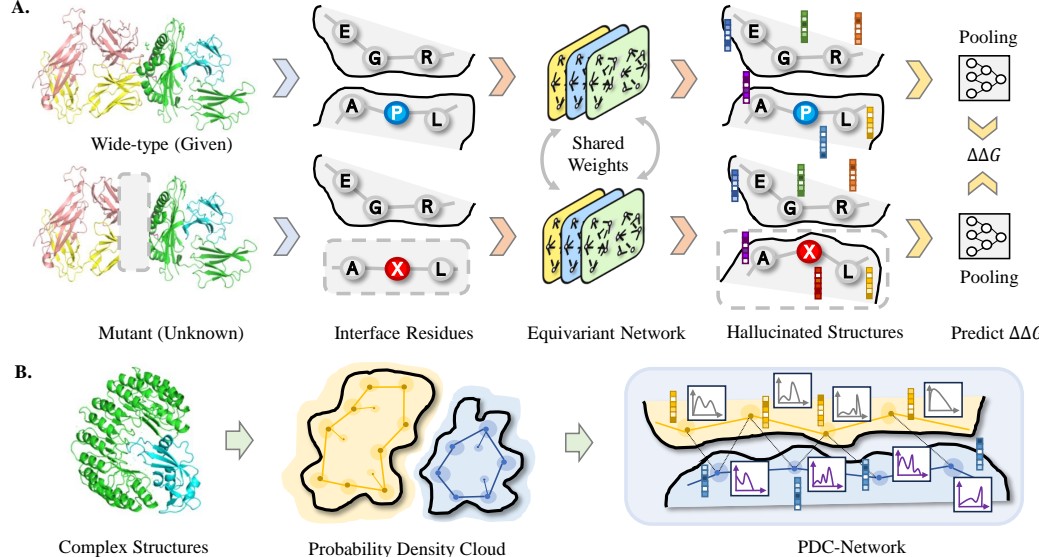

Figure 3: **A.** The overall pipeline of our Refine-PPI. The given wild-type structure and the masked mutant structure are subsequently fed into weight-shared equivariant neural networks. The masked region is reconstructed, and the mutation effect is predicted by comparing the features of two resulting complexes. **B.** The procedure of deep learning architecture. The particles in the complex are represented as probability density clouds (PDCs), where each atom moves according to some geometric distributions instead of being immobile. Then, the natural parameters including mean, variance, and co-variance are updated and propagated throughout PDC-Network.

roto-translational invariant features $\mathbf{h}_i \in \mathbb{R}^{\psi_0}$ (*e.g.*, atom or amino acid types, and electronegativity) and coordinates $\mathbf{x}_i \in \mathbb{R}^3$. $\mathcal{E}$ determines the connectivity between these particles and is divided into internal edges within each component as $\mathcal{E}_{\mathrm{L}}$ and $\mathcal{E}_{\mathrm{R}}$ and external edges between counterparts as $\mathcal{E}_{\mathrm{LR}}$. We assume $n$ residues in the entire complex and consistent residue numbers (*i.e.*, $\left|\mathcal{V}^{\mathrm{WT}}\right| = \left|\mathcal{V}^{\mathrm{MT}}\right| = n$). We select four backbone atoms $\{\mathrm{N}, \mathrm{C}_\alpha, \mathrm{C}, \mathrm{O}\}$ and an additional $\mathrm{C}_\beta$ to represent each amino acid.

**Problem Statement.** The mutation effect prediction is to approximate the ground-truth function that maps from the wild-type structure $\mathcal{G}^{\mathrm{WT}}$ and mutant information (*i.e.*, where and how some residues mutate from one type $a_i \in \{\mathrm{ACDEFGHIKLMNPQRSTVWY}\}$ to the other $a_i'$) to $\Delta\Delta G$.

## 3 METHOD

**Overview.** Refine-PPI (see Fig. 3) has three constituents parameterized by $\rho, \theta, \tau$, respectively. The backbone module $h_\rho(.)$ encodes the input 3D complex structure, the structure refinement module $f_\theta(.)$ hallucinates the unseen mutant structure, and the predictor $g_\tau(.)$ estimates the final $\Delta\Delta G$. The whole pipeline is described below. To begin with, the wild-type structure $\mathcal{G}^{\mathrm{WT}}$ and a well-initialized mutant structure $\tilde{\mathcal{G}}^{\mathrm{MT}}$ (the initialization details will be elucidated later) are fed into $h_\rho(.)$ to gain their corresponding features $\mathbf{Z}^{\mathrm{WT}} \in \mathbb{R}^{n \times \psi_1}$ and $\mathbf{Z}^{\tilde{\mathrm{MT}}} \in \mathbb{R}^{n \times \psi_1}$, respectively. Then, the imperfect mutant structure $\tilde{\mathcal{G}}^{\mathrm{MT}}$ along with its first-round representation $\mathbf{Z}^{\tilde{\mathrm{MT}}}$ is forwarded into $f_\theta(.)$ for several cycles and acquires the ultimate structure $\hat{\mathcal{G}}^{\mathrm{MT}}$ with more robust coordinates $\hat{\mathbf{x}}^{\mathrm{MT}}$. Subsequently, the hallucinated mutant structure $\hat{\mathcal{G}}^{\mathrm{MT}}$ is encoded by $h_\rho(.)$ again, and we can retrieve its second-round updated representation $\mathbf{Z}^{\mathrm{MT}} \in \mathbb{R}^{n \times \psi_1}$. As last, a pooling layer and $g_\tau(.)$ are appended to aggregate graph-level representations of both wild-type and mutation-type noted as $\mathbf{H}^{\mathrm{WT}} \in \mathbb{R}^{\psi_2}$ and $\mathbf{H}^{\mathrm{MT}} \in \mathbb{R}^{\psi_2}$ based on $\mathbf{Z}^{\mathrm{WT}}$ and $\mathbf{Z}^{\mathrm{MT}}$, and output the predicted free energy change $\hat{y}$.

**Mask Mutation Modeling.** As $\mathcal{G}^{\text{MT}}$ is hard to attain, we rely on the accessible $\mathcal{G}^{\text{WT}}$ to train $f_\theta(.)$ to restore the fragmentary structures. To this end, we introduce a mask mutation modeling (MMM) task, which requires $f_\theta(.)$ to reconstruct corrupted wild-type structures $\tilde{\mathcal{G}}^{\text{WT}}$. Here, we consider a single-mutation circumstance for better illustration where the $m$-th residue mutates from $a_m$ to $a'_m$. Then, a $(l+r)$-length segment around this mutation site is masked, denoted as $\mathcal{V}_{\text{mut}} = \{v_i\}_{i=m-l}^{m+r}$, which starts from the $(m-l)$-th residue and ends at the $(m+r)$-th residue. We aim to recover the structure of this masked region $\left\{\mathbf{x}^{\text{WT}}\right\}_{i=m-l}^{m+r}$ given $\tilde{\mathcal{G}}^{\text{WT}}$, its representation, and the native amino acid type $a_m$. The entire process is $f_\theta\left(\mathbf{Z}^{\tilde{\text{MT}}}, \tilde{\mathcal{G}}^{\text{WT}}, a_m\right) \to \left\{\mathbf{x}^{\text{WT}}\right\}_{i=m-l}^{m+r}$.

Intuitively, how to corrupt $\mathcal{G}^{\text{MT}}$ is significant, because the same corruption mechanism will be imposed to procure the incipient mutant structure $\tilde{\mathcal{G}}^{\text{MT}}$ during inference, serving as a starting point to deduce the final hallucinated structure $\hat{\mathcal{G}}^{\text{MT}}$. Here, we investigate two strategies to initialize coordinates of the masked regions $\mathcal{V}_{\text{mut}}$. Firstly, we borrow ideas from denoising-based molecular pretraining methods (Godwin et al., 2021; Feng et al., 2023) and independently add a random Gaussian noise of zero mean $\epsilon \sim \mathcal{N}(\mathbf{0}, \boldsymbol{\alpha})$ to the original coordinates as $\tilde{\mathbf{x}}_i^{\text{WT}} = \mathbf{x}_i^{\text{WT}} + \epsilon$, where $\boldsymbol{\alpha}$ determines the scale of the noisy deviation. This denoising objective is equivalent to learning a special force field (Zaidi et al., 2022).

In addition, we introduce a more challenging mode to corrupt $\mathcal{G}^{\text{MT}}$ and hypothesize that the mutant regions $\mathcal{V}_{\text{mut}}$ are completely unknown. To be specific, we initialize the coordinates the masked regions $\left\{\mathbf{x}^{\text{WT}}\right\}_{i=m-l}^{m+r}$ according to the even distribution between the residue right before the region (namely, $v_{m-l-1}$) and the residue right after the region (namely, $v_{m+r+1}$). Notably, residues immediately preceding or following the region can be missing, in which case we extend the existing side in reverse to initialize $\mathcal{V}_{\text{mut}}$ (see Fig. 7). The overall process is written as follows:

$$\tilde{\mathbf{x}}_i = \begin{cases} \mathbf{x}_{m-l-1} + (i-m+l+1)\frac{\mathbf{x}_{m+r+1}-\mathbf{x}_{m-l-1}}{l+r+2}, & \text{if } \exists v_{m-l-1}, v_{m+r+1}, \\ \mathbf{x}_{m+r+1} - (m+r+1-i)\left(\mathbf{x}_{m+r+2}-\mathbf{x}_{m+r+1}\right), & \text{if } \nexists v_{m-l-1}, \exists v_{m+r+1}, \\ \mathbf{x}_{m-l-1} + (i-m+l+1)\left(\mathbf{x}_{m-l-1}-\mathbf{x}_{m-l-2}\right), & \text{if } \exists v_{m-l-1}, \nexists v_{m+r+1}, \end{cases} \quad (1)$$

Noteworthily, both initialization strategies can be easily extended to multiple mutations.

After that, the corrupted wild-type structure $\tilde{\mathcal{G}}^{\text{WT}}$ is sent sequentially to $h_\rho(.)$ and $f_\theta(.)$ to restore the coordinates of the mutant regions masked, resulting in $\hat{\mathbf{x}}^{\text{WT}}$. As coordination data usually contains noise, we take the cue from MEAN (Kong et al., 2022) and adopt the Huber loss (Huber, 1992) instead of the common RMSD loss to avoid numerical instability. The loss function is defined by comparing to the actual $\mathbf{x}_i$:

$$\mathcal{L}_{\text{refine}} = \sum_{i \in \mathcal{V}_{\text{mut}}} \frac{1}{|\mathcal{V}_{\text{mut}}|} l_{\text{huber}}(\hat{\mathbf{x}}_i, \mathbf{x}_i). \quad (2)$$

$\Delta\Delta G$ **Prediction.** We impose the same strategy in MMM to initialize $\tilde{\mathcal{G}}^{\text{MT}}$ based on $\mathcal{G}^{\text{WT}}$. Then given the mutant information $a'_m$, we utilize weight-shared $h_\rho(.)$ and weight-shared $f_\theta(.)$ to hallucinate the unknown mutant structure as $p\left(\left\{\mathbf{x}^{\text{MT}}\right\}_{i=m-l}^{m+r} \middle| \tilde{\mathcal{G}}^{\text{MT}}, a'_m, \theta, \rho\right)$. It is worth noting that the resulting $\hat{\mathbf{x}}^{\text{WT}}$ does not carry gradients with no backpropagation at this phase. Later, we leverage $\mathcal{G}^{\text{WT}}$ and $\hat{\mathcal{G}}^{\text{MT}}$ to extract their corresponding representations $\mathbf{Z}^{\text{WT}}$ and $\mathbf{Z}^{\text{MT}}$, separately. $\mathbf{Z}^{\text{WT}}$ and $\mathbf{Z}^{\text{MT}}$ are then delivered to $g_\tau(.)$ to acquire the predicted change in free energy $\hat{y}$. Supervision is realized by the sum of two losses as $\mathcal{L} = \mathcal{L}_{\Delta\Delta G}(y, \hat{y}) + \lambda \mathcal{L}_{\text{refine}}\left(\left\{\mathbf{x}^{\text{WT}}\right\}_{i=m-l}^{m+r}, \left\{\hat{\mathbf{x}}^{\text{WT}}\right\}_{i=m-l}^{m+r}\right)$, where $\lambda$ is the balance hyperparameter. The whole paradigm illustrated in pseudo-code is put in Appendix 1.

**Discussion.** Previous studies exemplified by Google's DeepDream (Mordvintsev et al., 2015) train networks to recognize faces and other patterns in images, and invert and adjust arbitrary input images to draw more strongly resemble patterns perceived by the network. The generated images are often referred to as hallucinations because they may not faithfully represent any actual face, but what DL models view as an ideal face. This mechanism has also demonstrated success in macromolecules, where information stored in parameters of trained networks can be harnessed to design new protein structures featuring new sequences (Anishchenko et al., 2021). Refine-PPI uses a similar method to

explore whether networks trained on existing wild-type structures could be inverted to generate new 'ideal' protein structures based on mutant information. We discover that networks have the strong hallucination capacity to resolve the inevitable dilemma of the missing mutant structures.

### 3.1 PROBABILITY DENSITY CLOUD NETWORK

**Kinetics in Molecules.** Cutting-edge architectures extend networks to Euclidean and non-Euclidean domains, encompassing manifolds, meshes, or strings. As molecules can be naturally represented as graphs, graph approaches become dominant in molecular modeling (Schütt et al., 2018; Fuchs et al., 2020; Liao & Smidt, 2022). Beyond addressing GNNs' inherent limitations (Wu et al., 2022), they incorporate geometric principles like symmetry through equivariance and invariance. However, previous approaches were primarily designed for static and stable molecules characterized by deterministic and uncertainty-free structures. Here, we propose to integrate dynamics into geometric GNNs.

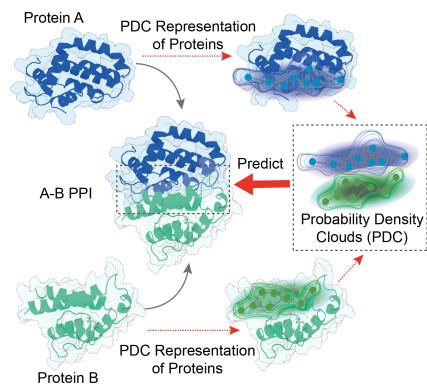

Figure 4: The PDC representations for a protein pair. Particles conform to geometric distributions like Gaussian with uncertainty in the 3D space.

**Probability Density Cloud.** Atoms are never at rest, even at extremely low temperatures (Clerk-Maxwell, 1873), and exhibit translational, rotational, or vibrational motion. In quantum mechanisms, electrons do not follow well-defined paths like planets around the Sun in classical physics but exist at specific energy levels and are described by wave functions, the mathematical functions on the probability of finding an electron in various locations around the nucleus (Schumaker, 1986). Physicists commonly envision an electron or other quantum particle by depicting their probability distribution around a specific region of space within an atom or molecule, where the shape and size of orbitals depend on the quantum numbers.

Inspired by this phenomenon, we portray particles as PDC showing regions with a higher probability of finding them. $\mathbf{x}_i$ are assumed to follow Gaussian as $\mathcal{N}(\boldsymbol{\mu}_i, \boldsymbol{\Sigma}_i)$. $\boldsymbol{\mu}_i \in \mathbb{R}^3$ is the place where node $i$ is most likely to be located, and $\boldsymbol{\Sigma}_i \in \mathbb{R}^{3 \times 3}$ is an isotropic (or spherical) covariance matrix signifying the independence upon the coordinate system. Given this premise, we can derive many invariant geometries that emphasize molecular structural information. The primary variable is the distance $d_{ij} = ||\mathbf{x}_i - \mathbf{x}_j||^2$. As $\mathbf{x}_i$ and $\mathbf{x}_j$ are are statistically independent, their difference follows a normal distribution as $\mathbf{x}_i - \mathbf{x}_j \sim \mathcal{N}(\boldsymbol{\mu}_i - \boldsymbol{\mu}_j, \boldsymbol{\Sigma}_i + \boldsymbol{\Sigma}_j)$ (Lemons, 2003), and its squared norm denoted as $d_{ij}^2$ exhibits a generalized chi-squared distribution $\chi^2(.)$ with a set of natural parameters, comprising $(\boldsymbol{\mu}_i - \boldsymbol{\mu}_j, \boldsymbol{\Sigma}_i + \boldsymbol{\Sigma}_j)$. The mean and variance of $\chi^2(.)$, denoted as $\mu_{d_{ij}}$ and $\sigma_{d_{ij}}$, are:

$$
\begin{aligned}
\mu_{d_{ij}} &= \operatorname{tr}(\boldsymbol{\Sigma}_i + \boldsymbol{\Sigma}_j) + ||\boldsymbol{\mu}_i - \boldsymbol{\mu}_j||^2, \\
\sigma_{d_{ij}} &= 2\operatorname{tr}(\boldsymbol{\Sigma}_i + \boldsymbol{\Sigma}_j) + 4(\boldsymbol{\mu}_i - \boldsymbol{\mu}_j)^\top (\boldsymbol{\Sigma}_i + \boldsymbol{\Sigma}_j)(\boldsymbol{\mu}_i - \boldsymbol{\mu}_j),
\end{aligned}
\tag{3}
$$

where $\operatorname{tr}(.)$ calculates the trace of a matrix. Furthermore, distributions of other geometric variables can also be induced. Let $\mathbf{x}_{ab}$ be the directed vector from $\mathbf{x}_a$ to $\mathbf{x}_b$, and consider triangle nodes $(i, j, k)$, the angle distribution $\angle \mathbf{x}_{ij} \mathbf{x}_{ik}$ can be characterized as the distribution of $\arccos \frac{(\mathbf{x}_i - \mathbf{x}_j) \cdot (\mathbf{x}_j - \mathbf{x}_k)}{|\mathbf{x}_i - \mathbf{x}_j||\mathbf{x}_j - \mathbf{x}_k|}$.

**PDC-Net.** Our PDC idea can be generalized to any geometric architecture and here we select EGNN (Satorras et al., 2021) as backbone. Our PDC-Net no longer accepts deterministic geometries $d_{ij}$ and $\mathbf{x}_i$, but takes distributions $f_{d_{ij}}$ and $f_{\mathbf{x}_i}$ as ingredients. Its $l$-th layer, named PDC-L, takes the set of node embeddings $\mathbf{h}^{(l)} = \left\{ \mathbf{h}_i^{(l)} \right\}_{i=1}^n$, edge information $\mathcal{E} = \{\mathcal{E}_L, \mathcal{E}_R, \mathcal{E}_{LR}\}$, and geometric feature distributions $\boldsymbol{\nu}^{(l)} = \left\{ \boldsymbol{\mu}_i^{(l)}, \boldsymbol{\Sigma}_i^{(l)} \right\}_{i=1}^n$ as input, and outputs a transformation on $\mathbf{h}^{(l+1)}$ and

$\boldsymbol{\nu}^{(l+1)}$. Concisely, $\mathbf{h}^{(l+1)}, \boldsymbol{\nu}^{(l+1)} = \text{PDC-L}\left[\mathbf{h}^{(l)}, \boldsymbol{\nu}^{(l)}, \mathcal{E}\right]$, which is defined as follows:

$$\mathbf{m}_{j\to i} = \phi_e\left(\mathbf{h}_i^{(l)}, \mathbf{h}_j^{(l)}, \mu_{d_{ij}}^{(l)}, \sigma_{d_{ij}}^{(l)}\right), \quad \mathbf{h}_i^{(l+1)} = \phi_h\left(\mathbf{h}_i^{(l)}, \sum_j \mathbf{m}_{j\to i},\right), \quad (4)$$

$$\boldsymbol{\mu}_i^{(l+1)} = \boldsymbol{\mu}_i^{(l)} + \frac{1}{|\mathcal{N}(i)|}\sum_{j\in\mathcal{N}(i)}\left(\boldsymbol{\mu}_i^{(l)} - \boldsymbol{\mu}_j^{(l)}\right)\phi_\mu(\mathbf{m}_{j\to i}), \quad (5)$$

$$\boldsymbol{\Sigma}_i^{(l+1)} = \boldsymbol{\Sigma}_i^{(l)} + \frac{1}{|\mathcal{N}(i)|}\sum_{j\in\mathcal{N}(i)}\left(\boldsymbol{\Sigma}_i^{(l)} + \boldsymbol{\Sigma}_j^{(l)}\right)\phi_\sigma(\mathbf{m}_{j\to i}), \quad (6)$$

where $\phi_e, \phi_h, \phi_\mu, \phi_\sigma$ are the edge, node, mean, and variance operations respectively that are commonly approximated by Multilayer Perceptrons (MLPs). It is worth noting that the mean position of each particle, denoted as $\boldsymbol{\mu}_i$, is updated through a weighted sum of all relative differences $(\boldsymbol{\mu}_i - \boldsymbol{\mu}_j)_{\forall j\in\mathcal{N}(i)}$. Meanwhile, the variance $\boldsymbol{\Sigma}_i$ is updated by a weighted sum of all additions $(\boldsymbol{\Sigma}_i + \boldsymbol{\Sigma}_j)_{\forall j\in\mathcal{N}(i)}$. These strategies align with the calculation of the mean and variance of the difference between two normal random variables. We also provide another type of mechanism to update the variance and observe a slight improvement in Appendix B.2. Regarding the initialization of $\boldsymbol{\Sigma}$, we explore three different approaches, and details are elucidated in the Appendix 4.3.1. Moreover, PDC-Net maintains the equivariance property, and the proof can be found in Appendix D.

## 4 RESULTS

### 4.1 EXPERIMENTAL SETUPS

**Data** Evaluation is carried out in SKEMPI.v2 (Jankauskaitė et al., 2019). It contains data on changes in the thermodynamic parameters and kinetic rate constants after mutation for structurally resolved PPIs. The latest version contains manually curated binding data for 7,085 mutations. The dataset is split into 3 folds by structure, each containing unique protein complexes that do not appear in other folds. Two folds are used for train and validation, and the remaining fold is used for test. This yields 3 different sets of parameters and ensures that every data point in SKEMPI.v2 is tested once. The pretraining data is derived from PDB-REDO, a database that contains refined X-ray structures in PDB. The protein chains are clustered based on 50% sequence identity, leading to 38,413 chain clusters, which are randomly divided into the training, validation, and test sets by 95%/0.5%/4.5% respectively.

**Baselines and Metrics.** We evaluate PDC-Net against various categories of techniques. The initial kind encompasses conventional empirical energy functions such as **Rossetta** Cartesian $\Delta\Delta G$ Park et al. (2016); Alford et al. (2017) and **FoldX**. The second grouping comprises sequence/evolution-based methodologies, exemplified by **ESM-1v** Meier et al. (2021), **PSSM**, **MSA Transformer** Rao et al. (2021), and Tranception Notin et al. (2022). The third category includes end-to-end learning models such as **DDGPred** Shan et al. (2022) and another **End-to-End** model that adopts Graph Transformer (GT) Luo et al. (2023) as the encoder architecture, but employs an MLP to directly forecast $\Delta\Delta G$. The fourth grouping encompasses unsupervised/semi-supervised learning approaches, consisting of **ESM-IF** Hsu et al. (2022) and MIF Yang et al. (2022). They pretrain networks on structural data and then employ the pretrained representations to predict $\Delta\Delta G$. MIF also utilizes GT as an encoder for comparative purposes with two variations: **MIF-$\Delta$logit** uses the disparity in log-probabilities of amino acid types to attain $\Delta\Delta G$, and **MIF-Network** predicts $\Delta\Delta G$ based on acquired representations. Besides, **B-factors** is the network that anticipates the B-factor of residues and incorporates the projected B-factor in lieu of entropy for $\Delta\Delta G$ prediction. Lastly, Rotamer Density Estimator (RDE) Luo et al. (2023) uses a flow-based generative model to estimate the probability distribution of rotamers and uses entropy to measure flexibility with two variants containing **RDE-Linear** and **RDE-Network**. **PPIFormer** (Bushuiev et al., 2023) is pretrained on a newly collected non-redundant 3D PPI interface dataset PPIRef through the mask language modeling (MLM) technique. More details are in the Appendix A.

Five metrics are used: Pearson and Spearman correlation coefficients, minimized RMSE, minimized MAE (mean absolute error), and AUROC (area under the receiver operating characteristic). Calculating AUROC involves classifying mutations according to the direction of their $\Delta\Delta G$ values. In

Table 1: Evaluation of $\Delta\Delta G$ prediction on the SKEMPI.v2 dataset.

| Method | Pretrain | Per-Structure | | Overall | | | | |
| | | Pearson | Spearman | Pearson | Spearman | RMSE | MAE | AUROC |
|---|---|---|---|---|---|---|---|---|
| **Energy Function-based** | | | | | | | | |
| Rosetta | – | 0.3284 | 0.2988 | 0.3113 | 0.3468 | 1.6173 | 1.1311 | 0.6562 |
| FoldX | – | 0.3789 | 0.3693 | 0.3120 | 0.4071 | 1.9080 | 1.3089 | 0.6582 |
| **Supervised-based** | | | | | | | | |
| DDGPred | ✗ | 0.3750 | 0.3407 | 0.6580 | 0.4687 | **1.4998** | **1.0821** | 0.6992 |
| End-to-End | ✗ | 0.3873 | 0.3587 | 0.6373 | 0.4882 | 1.6198 | 1.1761 | 0.7172 |
| **Sequence-based** | | | | | | | | |
| ESM-1v | ✓ | 0.0073 | -0.0118 | 0.1921 | 0.1572 | 1.9609 | 1.3683 | 0.5414 |
| PSSM | ✓ | 0.0826 | 0.0822 | 0.0159 | 0.0666 | 1.9978 | 1.3895 | 0.5260 |
| MSA Transf. | ✓ | 0.1031 | 0.0868 | 0.1173 | 0.1313 | 1.9835 | 1.3816 | 0.5768 |
| Tranception | ✓ | 0.1348 | 0.1236 | 0.1141 | 0.1402 | 2.0382 | 1.3883 | 0.5885 |
| **Unsupervised or Semi-supervised-based** | | | | | | | | |
| B-factor | ✓ | 0.2042 | 0.1686 | 0.2390 | 0.2625 | 2.0411 | 1.4402 | 0.6044 |
| ESM-IF | ✓ | 0.2241 | 0.2019 | 0.3194 | 0.2806 | 1.8860 | 1.2857 | 0.5899 |
| MIF-$\Delta$logit | ✓ | 0.1585 | 0.1166 | 0.2918 | 0.2192 | 1.9092 | 1.3301 | 0.5749 |
| MIF-Net. | ✓ | 0.3965 | 0.3509 | 0.6523 | 0.5134 | 1.5932 | 1.1469 | 0.7329 |
| RDE-Linear | ✓ | 0.2903 | 0.2632 | 0.4185 | 0.3514 | 1.7832 | 1.2159 | 0.6059 |
| RDE-Net. | ✓ | 0.4448 | 0.4010 | 0.6447 | 0.5584 | 1.5799 | 1.1123 | 0.7454 |
| PPIFormer | ✓ | 0.4281 | 0.3995 | 0.6450 | 0.5304 | 1.6420 | 1.1186 | 0.7380 |
| Refine-PPI | ✗ | 0.4475 | 0.4102 | 0.6584 | 0.5394 | 1.5556 | 1.0946 | 0.7517 |
| Refine-PPI | ✓ | **0.4561** | **0.4374** | **0.6592** | **0.5608** | 1.5643 | 1.1093 | **0.7542** |

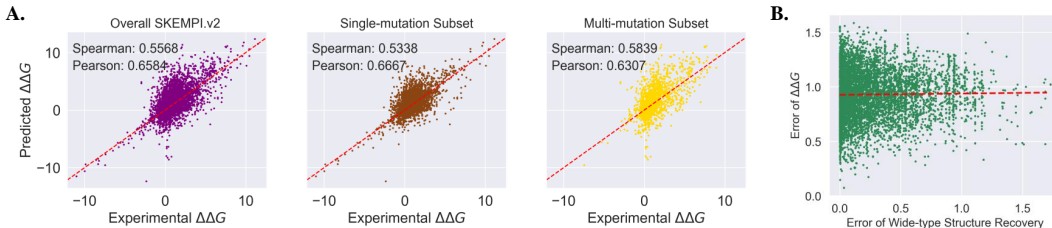

Figure 5: **A.** Visualization of correlations between experimental $\Delta\Delta G$ and predicted $\Delta\Delta G$. **D.** A selected example of the interface of a predicted mutant structure. **B.** The scatter plot shows that the recovery error of the wild-type structure has a positive relation with the error of $\Delta\Delta G$ prediction.

practical scenarios, the correlation observed within a specific protein complex attracts heightened interest. To account for this, we arrange mutations according to their associated structures. Groups with fewer than 10 mutation data points are excluded. Subsequently, correlation calculations are performed for each structure independently, leading to two additional metrics: **the average per-structure Pearson and Spearman correlation coefficients**. Other details are in the Appendix A.

## 4.2 COMPARISON WITH EXISTING TOOLS ON MUTANT EFFECT PREDICTION

Tab. 1 documents the results, and performance on subsets of single-mutation and multi-mutation is removed to Appendices 7 and 8 due to space limitation. It can be seen that our Refine-PPI model is better or more competitive in all regression metrics. Precisely, it achieves the highest per-structure Spearman and Pearson's correlations, which are considered as our primary metrics because the correlation of one specific protein complex is the most important.

Multiple point mutations are often required for successful affinity maturation (Sulea et al., 2018), and Refine-PPI outperforms DDGPred and RDE-Net by a large margin in the multi-mutation subset. This stems from the fact that RDE-Net and DDGPred perceive the mutant structures the same as the wild-type and consequently are not aware of the structural distinction. On the contrary, the mutant structures with multiple mutations should be more different than those with single mutations, and it becomes more crucial to detect the variant after the mutation. Refine-PPI anticipates the structural transformation due to mutation and can connect the structural change with $\Delta\Delta G$. Notably, Refine-

PPI trained from scratch has already outpassed pretrained methods such as RDE-Net, MIF-Net, and ESM-IF, which enjoy the unsupervised benefits in PDB-REDO. This further verifies the great success of Refine-PPI.

## 4.3 CORRELATION BETWEEN VARIANCE AND ATOMIC UNCERTAINTY

PDC posits that atoms adhere to Gaussian and derives geometric attributes such as distance and angles as distributions, where $\Sigma$ determines the magnitude of 3D atomic uncertainty. Here, we justify the correspondence between $\Sigma$ and positional uncertainty. Notably, experimentally observing and documenting particle uncertainty within macromolecules, such as proteins, is challenging. All data in PDB or SKEMPI.v2 are deterministic and uncertainty-free conformations. As a solution, we resort to molecular dynamics (MD) simulations to simulate atomic motions. Notably, MD approximates atomic motions by Newtonian physics and can capture the sequential behavior of molecules in full atomic details at a very fine temporal resolution. We run short-time MD for all complexes in SKEMPI.v2 and calculate the Root Mean Square Fluctuation (RMSF) alongside the entire trajectory, which numerically indicates positional differences between entire structures over time. It calculates individual residue flexibility, or how much a particular residue fluctuates during a simulation.

### 4.3.1 INITIALIZATION OF VARIANCE

We investigate three mechanisms to initialize $\Sigma$. First and naively, we turn all $\Sigma_i$ into an identity matrix $\mathbf{I}$. Second, we leverage RMSF as the initial $\Sigma$. Third, we adopt a learnable strategy to initialize $\Sigma$, where an embedding layer maps each category of twenty residue types to their corresponding $\Sigma$. Their performance is listed

Table 2: Performance of different initialization methods for the coordinate variance $\Sigma$ (without pretraining)

| Method | Per-Structure | |
|---|---|---|
| | Pearson | Spearman |
| Identity Matrix | $0.4422 \pm 0.0033$ | $0.4043 \pm 0.0018$ |
| MD Simulations | $\mathbf{0.4522} \pm 0.0036$ | $\mathbf{0.4287} \pm 0.0015$ |
| Learnable $\Sigma$ | $0.4475 \pm 0.0034$ | $0.4102 \pm 0.0017$ |

in Tab. 2, where the mean and standard deviation are documented for three runs. It can be found that MD-based initialization achieves the best Spearman (0.4287), outweighing the learnable one (0.4102) and the identity matrix (0.4043), emphasizing the efficacy of incorporating simulated uncertainty into the PDC module. This implies that simulated uncertainty is the optimal choice for this variance, and learned variance ideally should move towards this simulated uncertainty. However, since MD simulations are time-consuming and costly, it is prohibited to implement MD during the inference stage each time. As a consequence, we use the learnable sort in Refine-PPI for subsequent experiments.

### 4.3.2 ANALYSIS OF LEARNED UNCERTAINTY

**Visualization of Learned Variance.** We randomly pick up four PDBs and visualize the learned variance, that is, the magnitude of $||\Sigma_i||^2$ in Fig. 5. Pictures show that particles at the interface have a smaller variation than those at protein edges. This aligns with the biological concept that atoms in the binding surface are less volatile than atoms in other parts of the complex. This phenomenon confirms that PDC-Net has adaptively comprehended the magnitude and strength of entities' motion during PPIs.

**Quantitative Analysis.** We also quantitatively investigated the correlation between the learned variance and the ground truth uncertainty. A detailed comparison, classified by residues at and not at the interface, is in Tab. 3. Notably, the ground truth RMSF at the interface is signif-

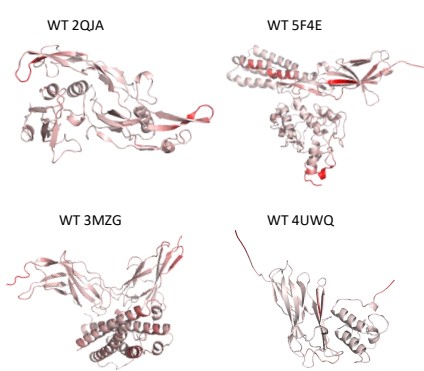

Figure 6: Visualization of learned uncertainty. A darker color corresponds to a more flexible protein segment.

icantly smaller than that observed elsewhere. At the same time, the learned $\Sigma_i$ exhibits a parallel pattern, where $||\Sigma_i||^2$ at the interface is much smaller. This analysis further substantiates that the learned variance corresponds to atomic uncertainty.

Table 3: Performance of different position variance update methods without pretraining.

|  | Interface | Non-Interface | Overall |
|---|---|---|---|
| RMSF | 0.4945 | **0.9735** | 0.8271 |
| $\|\mathbf{\Sigma}_i\|^2$ | 0.6072 | **0.8940** | 0.7745 |

### 4.3.3 PERFORMANCE OF UNCERTAINTY PREDICTION WITH MD SIMULATIONS

To further verify the efficacy of our PDC-Net to capture the atomic uncertainty, we propose a more straightforward task, where DL models are required to directly predict the simulated uncertainty (*i.e.*, RMSF). On the one hand, we adopt PDC-EGNN and directly enforce the learnable variance to correspond to the simulated uncertainty. The loss term is therefore set as $\mathrm{MSE}(\|\mathbf{\Sigma}_i\|^2, \mathrm{RMSF})$. On the other hand, we leverage some advanced geometric networks and require them to output RMSF based on the residue feature of the final layer. The loss is written as $\mathrm{MSE}(\mathrm{MLP}(\mathbf{h}^{(L)}), \mathrm{RMSF})$, where MLP is the abbreviation of the multi-layer perceptron. A group of baselines

Table 4: Performance on the uncertainty prediction task.

| Model | MSE |
|---|---|
| SchNet | $0.5214 \pm 0.038$ |
| GVP-GNN | $0.2807 \pm 0.025$ |
| SE(3)-Trans. | $0.3462 \pm 0.035$ |
| EGNN | $0.2609 \pm 0.026$ |
| TorchMD-Net | $0.2011 \pm 0.013$ |
| SphereNet | $0.1688 \pm 0.021$ |
| EquiFormer | $0.1659 \pm 0.022$ |
| PDC-EGNN | $\mathbf{0.1381 \pm 0.020}$ |

are selected for thorough comparison, including SchNet (Schütt et al., 2018), GVP-GNN (Jing et al., 2020), SE(3)-Transformer (Fuchs et al., 2020), SphereNet (Liu et al., 2021), TorchMD-Net (Thölke & De Fabritiis, 2021), and EquiFormer (Liao & Smidt, 2022). We run three random seeds and report the mean and standard deviation of these three runs in Tab. 4. The experiments show that the PDC module achieves the best performance in understanding the atomic uncertainty and significantly improves the ability of EGNN to forecast RMSF. This phenomenon illustrates that our design of $\mathbf{\Sigma}$ can be a good choice to represent and encode atomic uncertainty in the 3D space.

To summarize, though our loss term primarily influences output positions without directly enforcing the network to capture uncertainty information, extensive experiments demonstrate that the theoretical foundation of our PDC-module design closely connects the concept of atomic uncertainty with the variance of positional distributions $\mathbf{\Sigma}$.

### 4.4 DISCUSSION OF REFINE-PPI

**Ablation Studies.** We also conduct additional experiments to investigate the contributions of each component of our Refine-PPI and the results are displayed in Tab. 6. It can be concluded that the introduction of co-training of the structure refinement and the $\Delta\Delta G$ prediction greatly contributes to the promotion of all metrics, culminating in an increase of $11.8\%$ and $15.6\%$ in per-structure Pearson's and Spearman correlations. Additionally, PDC-Net

Table 5: Performance of different coordinate initialization strategies for MMM.

| Method | Per-Structure | |
|---|---|---|
|  | Pearson | Spearman |
| Easy | 0.4417 | 0.4060 |
| Hard | **0.4475** | **0.4102** |

also brings obvious benefits such as a lower MAE and a higher AUORC. In Tab. 5, we report the performance of two initialization strategies to corrupt the masked region. The easy mode (denoising-based) is slightly outpassed by the hard one (surroundings-based).

Table 6: Ablation study of Refine-PPI without pretraining, where we choose the backbone $h_\rho$ (*i.e.*, Graph Transformer) as the foundation model for comparison (*i.e.*, No. 1).

| No. | MMM | PDC-Net | Per-Structure | | Overall | | | | |
|---|---|---|---|---|---|---|---|---|---|
|  |  |  | Pearson | Spearman | Pearson | Spearman | RMSE | MAE | AUROC |
| 1 | ✗ | ✗ | 0.3708 | 0.3353 | 0.6210 | 0.4907 | 1.6199 | 1.1933 | 0.7225 |
| 2 | ✓ | ✗ | 0.4145 | 0.3875 | 0.6571 | **0.5553** | 1.5580 | 1.1025 | 0.7460 |
| 3 | ✓ | ✓ | **0.4475** | **0.4102** | **0.6584** | 0.5394 | **1.5556** | **1.0946** | **0.7517** |

**Visualization of Results.** We envision the scatter plot of experimental and predicted $\Delta\Delta G$ and also draw the relation between the error of wild-type structure recovery and the error of $\Delta\Delta G$ esti-

mation in Fig. 5. It can be found that, generally, a small error of wild-type structure reconstruction leads to a more accurate $\Delta\Delta G$ prediction. This indicates that these two tasks are closely related to each other. In addition, we provide a case study of 16 seed complexes with different numbers of mutations that are well predicted by our Refine-PPI in Fig. 9. It can be discovered that Refine-PPI can realize a pretty high Spearman of 0.7 even when there are more than three mutations. In addition, we visualize three hallucinated mutant structures in the Appendix C.

## 5 CONCLUSION

This work proposes a new framework named Refine-PPI to predict the mutation effect. Given that mutant structures are always absent, we introduce an additional structure refinement module to recover the masked regions around the mutations. This module is trained simultaneously via mask geometric modeling. In addition, we notice that protein-protein interactions are a dynamic process, but few prior studies have taken this characteristic into account in a deep learning design. To bridge the gap, we present a probability density cloud (PDC)-Network to capture the dynamics in atomic resolution. Our results highlight the necessity to adopt a more robust mutant structure and consider dynamics for molecular modeling. A statement regarding limitations and future work is elaborated in Appendix E.

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

## A  EXPERIMENTAL DETAILS

We implement all experiments on 4 A100 GPUs, each with 80G memory. Refine-PPI is trained with an Adam optimizer without weight decay and with $\beta_1 = 0.9$ and $\beta_2 = 0.999$. A ReduceL-ROnPlateau scheduler is employed to automatically adjust the learning rate with a patience of 10 epochs and a minimum learning rate of $1.e - 6$. The batch size is set to 64 and an initial learning rate of $1.e - 4$. The maximum iterations are 50K and the validation frequency is 1K iterations. The node dimension is 128, and no dropout is conducted. As for the structure refinement, the recycle number is set as 3, and the balance weight is tuned as 1.0. We performed a grid search to find the optimal length of the masked region and found that $l = r = 5$ is a good choice. However, different initializations require different optimal hyperparameters, and typically we can mask longer regions for denoising-based MMM. The pretraining follows a similar training scheme with a batch size of 32. During pretraining, the data loader randomly selects a cluster and then randomly chooses a chain from the cluster to ensure balanced sampling. Since there is no mutant residue in PDB-REDO, we randomly select a seed residue from the chosen chain and adopt the same MMM strategy.

As for the specific model architecture, the backbone module $h_\rho(.)$ can take the form of any conventional geometric neural network (e.g., GVP-GNN, EGNN, SE(3)-Transformer, Graph Transformer). Here, we adopt a one-layer Graph Transformer (Luo et al., 2023) to extract general representations of proteins. The refinement module $f_\theta(.)$ needs to output both updated features and coordinates, and therefore we use PDC-EGNN as $f_\theta(.)$ in our experiments. Lastly, the head predictor $g_\tau(.)$ is a simple linear layer that accepts the concatenation of representations of both wide and mutation types and forecasts the change in free energy. The total model size of our Refine-PPI is approximately 6M.

### A.1  BASELINES IMPLEMENTATIONS

Baselines that require training and calibration using the SKEMPI.v2 dataset (DDGPred, End-to-End, B-factor, MIF-$\Delta$logit, MIF-Network, RDE-Linear, and RDE-Net) are trained independently using the 3 different splits of the dataset as described in Section 4.1. This is to ensure that every data point in the SKEMPI.v2 dataset is tested once. Below are descriptions of the implementation of the baseline methods, which follow the same scheme as Luo et al. (2023) and Bushuiev et al. (2023).

**Rosetta** (Alford et al., 2017; Leman et al., 2020): The Rosetta version is 2021.16, and the scoring function is ref2015_cart. Every protein structure in the SKEMPI.v2 dataset is first pre-processed using the relax application. The mutant structure is built by cartesian_ddg. The binding free energies of both wild-type and mutant structures are predicted by interface_energy (dG_separated/dSASAx100). Finally, the binding $\Delta\Delta G$ is calculated by subtracting the binding energy of the wild-type structure from the binding energy of the mutant.

**FoldX** (Delgado et al., 2019): Structures are first relaxed by the RepairPDB command. Mutant structures are built with the BuildModel command based on the repaired structure. The change in binding free energy $\Delta\Delta G$ is calculated by subtracting the wild-type energy from the mutant energy.

**ESM-1v** (Meier et al., 2021): We use the implementation provided in the ESM open-source repository. Protein language models can only predict the effect of mutations for single protein sequences. Therefore, the cases where mutations occur in multiple sequences are ignored. The sequence of the mutated protein chain is extracted from the SEQRES entry in the PDB file. A masked marginal mode is used to score both wild-type and mutant sequences and use their difference as an estimate of $\Delta\Delta G$.

**PSSM** MSAs are constructed from the Uniref90 database for chains with mutation annotations in the SKEMPI.v2 dataset. Jackhmmer version 3.3.1 is used following the setting in Meier et al. (2021). The MSAs are filtered using HHfilter with coverage 75 and sequence identity 90. This HHfilter parameter is reported to have the best performance for MSA Transformer according to Meier et al. (2021). Position-specific scoring matrices (PSSM) is calculated and the change in probability is used as a prediction of $\Delta\Delta G$.

**MSA Transformer** (Rao et al., 2021): We use the implementation provided in the ESM open-source repository. We input the MSAs constructed during the evaluation of the PSSM to the MSA Transformer. We used the mask-marginal mode to score both wild-type and mutant sequences and use their difference as the prediction of $\Delta\Delta G$.

**Tranception** (Notin et al., 2022): We use the implementation provided in the Tranception open-source repository. We predict mutation effects using the large model checkpoint. Previously built MSAs (not filtered by HHfilter) are used for inference-time retrieval.

**DDGPred** (Shan et al., 2022): We use the implementation that follows the paper by Shan et al. (2022). Since this model requires predicted sidechain structures of the mutant, we use mutant structures packed during our evaluation of Rosetta to train the model and run prediction.

**End-to-End**: The end-to-end model shares the same encoder architecture as RDE (Luo et al., 2023). The difference is that in the RDE normalizing flows follow the encoder to model rotamer distributions, but in the end-to-end model, the embeddings are directly fed to an MLP to predict $\Delta\Delta G$.

**B-factor**: This model predicts per-atom b-factors for proteins. It has the same encoder architecture as RDE (Luo et al., 2023). The encoder is followed by an MLP that predicts a vector for each amino acid, where each dimension is the predicted b-factor of different atoms in the amino acid. The amino acid-level b-factor is calculated by averaging the atom-level b-factors. The predicted b-factors are used as a measurement of conformational flexibility. They are used to predict $\Delta\Delta G$ using the linear model same as RDE-Linear (Luo et al., 2023).

**ESM-IF** (Hsu et al., 2022): ESM-IF can score protein sequences using the log-likelihood. Implementation of the scoring function is provided in the ESM repository. We enable the –multichain_backbone flag to let the model see the whole protein-protein complex. We subtract the log-likelihood of the wild-type from the mutant to predict $\Delta\Delta G$.

**MIF Architecture**: The masked inverse folding (MIF) network uses the same encoder architecture as RDE (Luo et al., 2023). Following the encoder is a per-amino-acid 20-category classifier that predicts the type of masked amino acids. We use the same PDB-REDO train-test split to train the model. At training time, we randomly crop a patch consisting of 128 residues and randomly mask 10% amino acids. The model learns to recover the type of masked amino acids with the standard cross entropy loss.

**MIF-$\Delta$logit**: To score mutations, we first mask the type of mutated amino acids. Then, we use the log probability of the amino acid type as the score. Analogously, we have the score of the wild-type bound ligand, wild-type bound receptor, wild-type unbound ligand, unbound receptor, mutated bound ligand, mutated bound receptor, and mutated unbound ligand. Therefore, we use the identical linear model to RDE-Linear (Luo et al., 2023) to predict $\Delta\Delta G$ from the scores.

**MIF-Network**: This is similar to RDE-Network (Luo et al., 2023). The difference is that we use the pre-trained encoder of MIF rather than the encoder of RDE. We also freeze the MIF encoder as we aim to utilize the unsupervised representations.

**PPIFormer**: We use EquiFormer as the backbone and pretrain it on PPIRef. Then the effects of mutations are predicted via the log odds ratio.

## A.2 VISUALIZATION OF COORDINATE INITIALIZATION IN MMM

To better clarify the initialization of our MMM, we show the process of two different mechanisms (*i.e.*, the easy denoising-based one and the hard surrounding-based one) in Fig. 7.

## A.3 PSEUDO-CODE OF REFINE-PPI

# B ADDITIONAL RESULTS

## B.1 PERFORMANCE ON SUBSETS AND CASE STUDIES

For better comparison of our Refine-PPI and other baselines, we make a bar plot on per-structure Pearson's and Spearman correlations in Fig. 8. We also explicitly document the evaluation results of different methods on the multi-mutation and single-mutation subsets of the SKEMPI.v2 dataset in Tab. 7 and Tab. 8. It can be found that with pretraining on PDB-REDO, Refine-PPI achieves the best per-structure metrics on both multi-mutation and single-mutation subsets. This indicates that Refine-PPI is a more effective tool to screen and select mutant proteins for desired properties.

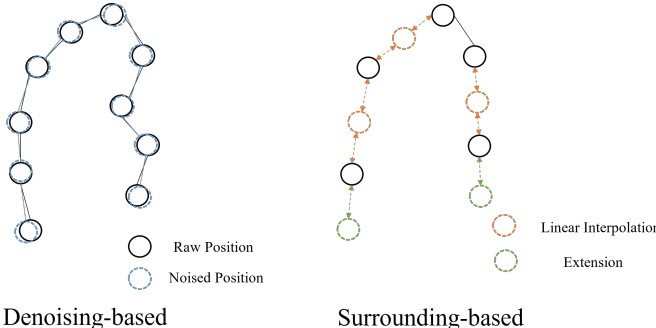

Figure 7: The illustration of coordinate initialization in the MMM task.

---

**Algorithm 1** The workflow of our Refine-PPI.

---

**Input:** wild-type structure $\mathcal{G}^{\text{WT}}$, mutant site and amino acid types $a_m$ and $a'_m$; backbone module $h_\rho$, refinement model $f_\theta$, head predictor $g_\tau$; number of recycles $k$, the real free energy change $y$, loss weight $\lambda$

$\tilde{\mathcal{G}}_0^{\text{WT}}, \tilde{\mathcal{G}}_0^{\text{MT}} \leftarrow$ Equation 1 $\left(\mathcal{G}^{\text{WT}}\right)$      ▷ Initialize structures

\# Training-only

**for** $t = 0, 1, ..., k-1$ **do**

    $\mathbf{Z}_t^{\text{WT}} \leftarrow h_\rho \left(\tilde{\mathcal{G}}_t^{\text{WT}}\right)$

    $\tilde{\mathbf{x}}_{t+1}^{\text{WT}} \leftarrow f_\theta \left(\tilde{\mathcal{G}}_t^{\text{WT}}, \mathbf{Z}_t^{\text{WT}}, \tilde{\mathbf{x}}_t^{\text{WT}}, a_m\right)$

**end for**

$\mathcal{L}_{\text{refine}} \leftarrow$ Equation 2 $\left(\tilde{\mathbf{x}}_k^{\text{WT}}, \mathbf{x}^{\text{WT}}\right)$      ▷ The MMM loss

**for** $t = 0, 1, ..., k-1$ **do**

    $\mathbf{Z}_t^{\text{MT}} \xleftarrow{\text{No grad.}} h_\rho \left(\tilde{\mathcal{G}}_t^{\text{MT}}\right)$

    $\tilde{\mathbf{x}}_{t+1}^{\text{MT}} \xleftarrow{\text{No grad.}} f_\theta \left(\tilde{\mathcal{G}}_t^{\text{MT}}, \mathbf{Z}_t^{\text{MT}}, \tilde{\mathbf{x}}_t^{\text{MT}}, a'_m\right)$

**end for**

$\mathbf{Z}^{\text{WT}}, \mathbf{Z}^{\text{MT}} \leftarrow h_\rho \left(\mathcal{G}^{\text{WT}}\right), h_\rho \left(\tilde{\mathcal{G}}_k^{\text{MT}}\right)$

$\hat{y} \leftarrow g_\tau \left(\mathbf{Z}^{\text{WT}}, \mathbf{Z}^{\text{MT}}\right)$

$\mathcal{L}_{\Delta\Delta G} \leftarrow$ RMSE$(\hat{y}, y)$      ▷ The $\Delta\Delta G$ loss

\# Backpropagation

$\rho, \theta, \tau \leftarrow \mathcal{L}_{\Delta\Delta G} + \lambda \mathcal{L}_{\text{refine}}$

---

### B.2 POSITION VARIANCE UPDATE IN PDC-EGNN

Notably, the way to update the variance of the positions of different atoms is not unique. Here, we offer another kind of approach to renew the variance in the layer of PDC-EGNN.

$$\mathbf{\Sigma}_i^{(l+1)} = \left(1 + \frac{1}{|\mathcal{N}(i)|} \sum_{j \in \mathcal{N}(i)} \phi_\mu(\mathbf{m}_{j \to i})\right)^2 \mathbf{\Sigma}_i^{(l)} + \frac{1}{|\mathcal{N}(i)|} \sum_{j \in \mathcal{N}(i)} \phi_\mu(\mathbf{m}_{j \to i})\mathbf{\Sigma}_j^{(l)}, \qquad (7)$$

where we leverage the same $\phi_\mu$ instead of a new $\phi_\sigma$. Besides, we distribute and square the $\mathbf{x}_i$ terms because $\mathbf{x}_i - \mathbf{x}_j$ is not independent of $\mathbf{x}_i$. Noticeably, this Equation 7 does not damage the equivariance property of our model. Experiments show that this form of position variance computation performs slightly better in the mutant effect prediction task (see Tab. 9), with a per-structure Spearman of 0.4490.

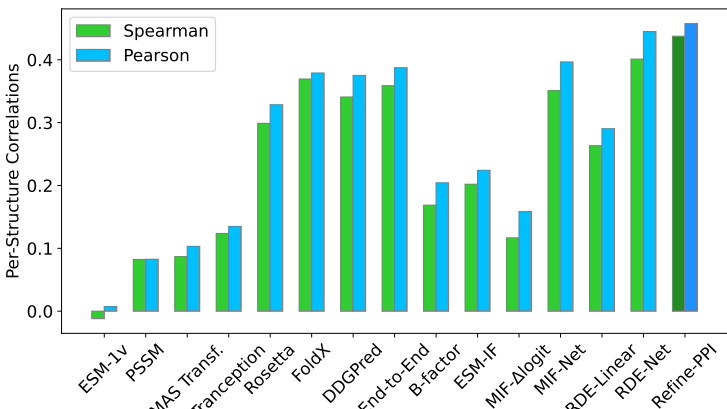

Figure 8: Per-structure Spearman and Pearson correlations of different baseline methods and Refine-PPI.

Table 7: Evaluation of $\Delta\Delta G$ prediction on the multi-mutation subset of the SKEMPI.v2 dataset.

| Method | Pretrain | Per-Structure | | Overall | | | | |
| | | Pearson | Spearman | Pearson | Spearman | RMSE | MAE | AUROC |
|---|---|---|---|---|---|---|---|---|
| **Energy Function-based** | | | | | | | | |
| Rosetta | – | 0.1915 | 0.0836 | 0.1991 | 0.2303 | 2.6581 | 2.0246 | 0.6207 |
| FoldX | – | 0.3908 | 0.3640 | 0.3560 | 0.3511 | 1.5576 | 1.0713 | 0.6478 |
| **Supervised-based** | | | | | | | | |
| DDGPred | ✗ | 0.3912 | 0.3896 | 0.5938 | 0.5150 | 2.1813 | 1.6699 | 0.7590 |
| End-to-End | ✗ | 0.4178 | 0.4034 | 0.5858 | 0.4942 | 2.1971 | 1.7087 | 0.7532 |
| **Sequence-based** | | | | | | | | |
| ESM-1v | ✓ | -0.0599 | -0.1284 | 0.1923 | 0.1749 | 2.7586 | 2.1193 | 0.5415 |
| PSSM | ✓ | -0.0174 | -0.0504 | -0.1126 | -0.0458 | 2.7937 | 2.1499 | 0.4442 |
| MSA Transf. | ✓ | -0.0097 | -0.0400 | 0.0067 | 0.0030 | 2.8115 | 2.1591 | 0.4870 |
| Tranception | ✓ | -0.0688 | -0.0120 | -0.0185 | -0.0184 | 2.9280 | 2.2359 | 0.4874 |
| **Unsupervised or Semi-supervised-based** | | | | | | | | |
| B-factor | ✓ | 0.2078 | 0.1850 | 0.2009 | 0.2445 | 2.6557 | 2.0186 | 0.5876 |
| ESM-IF | ✓ | 0.2016 | 0.1491 | 0.3260 | 0.3353 | 2.6446 | 1.9555 | 0.6373 |
| MIF-$\Delta$logit | ✓ | 0.1053 | 0.0783 | 0.3358 | 0.2886 | 2.5361 | 1.8967 | 0.6066 |
| MIF-Net. | ✓ | 0.3968 | 0.3789 | 0.6139 | 0.5370 | 2.1399 | 1.6422 | 0.7735 |
| RDE-Linear | ✓ | 0.1763 | 0.2056 | 0.4583 | 0.4247 | 2.4460 | 1.8128 | 0.6573 |
| RDE-Net. | ✓ | 0.4233 | 0.3926 | 0.6288 | 0.5900 | 2.0980 | 1.5747 | 0.7749 |
| PPIFormer | ✓ | 0.3985 | 0.3925 | 0.6405 | 0.5946 | 2.1407 | 1.5753 | 0.7893 |
| Refine-PPI | ✗ | 0.4474 | 0.4134 | 0.6307 | 0.5839 | 2.0939 | 1.5894 | 0.7831 |
| Refine-PPI | ✗ | **0.4558** | **0.4289** | **0.6458** | **0.6091** | **2.0601** | **1.554** | **0.8064** |

## C  VISUALIZATION OF HALLUCINATED STRUCTURES

Here we provide some instances of mutant structures hallucinated by our Refine-PPI in Fig. 10. Since the ground truth mutant structures are inaccessible, we leave it for future work to examine their accuracy.

## D  PROOF OF EQUIVARIANCE

Equivariance is an important characteristic, and here, we demonstrate that PDC-Net strictly follows this rule of principle. More formally, for any translation vector $g \in \mathbb{R}^3$ and for any orthogonal matrix $Q \in \mathbb{R}^{3\times3}$, the model should satisfy:

$$\mathbf{h}^{(l+1)}, \left\{Q\boldsymbol{\mu}_i^{(l+1)} + g, Q^\top\boldsymbol{\Sigma}_i^{(l+1)}Q\right\}_{i=1}^n = \text{PDC-L}\left[\mathbf{h}^{(l)}, \left\{Q\boldsymbol{\mu}_i^{(l)} + g, Q^\top\boldsymbol{\Sigma}_i^{(l)}Q\right\}_{i=1}^n, \mathcal{E}\right]. \quad (8)$$

Table 8: Evaluation of $\Delta\Delta G$ prediction on the single-mutation subset of the SKEMPI.v2 dataset.

| Method | Pretrain | Per-Structure | | Overall | | | | |
| | | Pearson | Spearman | Pearson | Spearman | RMSE | MAE | AUROC |
|---|---|---|---|---|---|---|---|---|
| **Energy Function-based** | | | | | | | | |
| Rosetta | – | 0.3284 | 0.2988 | 0.3113 | 0.3468 | 1.6173 | 1.1311 | 0.6562 |
| FoldX | – | 0.3908 | 0.3640 | 0.3560 | 0.3511 | 1.5576 | 1.0713 | 0.6478 |
| **Supervised-based** | | | | | | | | |
| DDGPred | ✗ | 0.3711 | 0.3427 | 0.6515 | 0.4390 | 1.3285 | 0.9618 | 0.6858 |
| End-to-End | ✗ | 0.3818 | 0.3426 | 0.6605 | 0.4594 | 1.3148 | 0.9569 | 0.7019 |
| **Sequence-based** | | | | | | | | |
| ESM-1v | ✓ | 0.0422 | 0.0273 | 0.1914 | 0.1572 | 1.7226 | 1.1917 | 0.5492 |
| PSSM | ✓ | 0.1215 | 0.1229 | 0.1224 | 0.0997 | 1.7420 | 1.2055 | 0.5659 |
| MSA Transf. | ✓ | 0.1415 | 0.1293 | 0.1755 | 0.1749 | 1.7294 | 1.1942 | 0.5917 |
| Tranception | ✓ | 0.1912 | 0.1816 | 0.1871 | 0.1987 | 1.7455 | 1.1708 | 0.6089 |
| **Unsupervised or Semi-supervised-based** | | | | | | | | |
| B-factor | ✓ | 0.1884 | 0.1661 | 0.1748 | 0.2054 | 1.7242 | 1.1889 | 0.6100 |
| ESM-IF | ✓ | 0.2308 | 0.2090 | 0.2957 | 0.2866 | 1.6728 | 1.1372 | 0.6051 |
| MIF-$\Delta$logit | ✓ | 0.1616 | 0.1231 | 0.2548 | 0.1927 | 1.6928 | 1.1671 | 0.5630 |
| MIF-Net. | ✓ | 0.3952 | 0.3479 | 0.6667 | 0.4802 | 1.3052 | 0.9411 | 0.7175 |
| RDE-Linear | ✓ | 0.3192 | 0.2837 | 0.3796 | 0.3394 | 1.5997 | 1.0805 | 0.6027 |
| RDE-Net. | ✓ | 0.4687 | 0.4333 | 0.6421 | 0.5271 | 1.3333 | 0.9392 | 0.7367 |
| PPIFormer | ✓ | 0.4192 | 0.3796 | 0.6287 | 0.4772 | 1.4232 | 0.9562 | 0.7213 |
| Refine-PPI | ✗ | 0.4474 | 0.4134 | **0.6667** | **0.5338** | **1.2963** | **0.9179** | 0.7431 |
| Refine-PPI | ✓ | **0.4701** | **0.4459** | 0.6658 | 0.5153 | 1.2978 | 0.9287 | **0.7481** |

Table 9: Performance of different position variance update methods (without pretraining).

| Method | Per-Structure | |
| | Pearson | Spearman |
|---|---|---|
| Equ. 6 | 0.4475 | 0.4102 |
| Equ. 7 | **0.4490** | **0.4153** |

We will analyze how the translation and rotation of input coordinates propagate through our model. We start by assuming that $\mathbf{h}^0$ is invariant to the $\mathrm{E}(n)$ transformations on the coordinate distributions $\boldsymbol{\nu}$. In other words, information on the absolute position or orientation of $\boldsymbol{\nu}^0$ is not encoded in $\mathbf{h}^0$. Then, the distance between two particles is invariant to translations, rotations, and reflections. This is because, for the mean of distance $\mu_{d_{ij}}$, we have $\mathrm{tr}\left(Q^\top \boldsymbol{\Sigma}_i Q + Q^\top \boldsymbol{\Sigma}_j Q\right) = \mathrm{tr}\left(\boldsymbol{\Sigma}_i + \boldsymbol{\Sigma}_j\right)$ due to the characteristic of the isotropic matrix and $\|Q\boldsymbol{\mu}_i^{(l)} + g - (Q\boldsymbol{\mu}_j^{(l)} + g)\|^2 = \|Q\boldsymbol{\mu}_i^{(l)} - Q\boldsymbol{\mu}_j^{(l)}\|^2 = (\boldsymbol{\mu}_i^{(l)} - \boldsymbol{\mu}_j^{(l)})^\top Q^\top Q(\boldsymbol{\mu}_i^{(l)} - \boldsymbol{\mu}_j^{(l)}) = (\boldsymbol{\mu}_i^{(l)} - \boldsymbol{\mu}_j^{(l)})^\top \mathbf{I}(\boldsymbol{\mu}_i^{(l)} - \boldsymbol{\mu}_j^{(l)}) = \|\boldsymbol{\mu}_i^{(l)} - \boldsymbol{\mu}_j^{(l)}\|^2$. Meanwhile, for the variance of distance $\sigma_{d_{ij}}$, we have $[Q\boldsymbol{\mu}_i + g - (Q\boldsymbol{\mu}_j + g)]^\top \left(Q^\top \boldsymbol{\Sigma}_i Q + Q^\top \boldsymbol{\Sigma}_j Q\right)[Q\boldsymbol{\mu}_i + g - (Q\boldsymbol{\mu}_j + g)] = (\boldsymbol{\mu}_i - \boldsymbol{\mu}_j)^\top Q^\top \left(\boldsymbol{\Sigma}_i + \boldsymbol{\Sigma}_j\right) Q(\boldsymbol{\mu}_i - \boldsymbol{\mu}_j) = (\boldsymbol{\mu}_i - \boldsymbol{\mu}_j)^\top \left(\boldsymbol{\Sigma}_i + \boldsymbol{\Sigma}_j\right)(\boldsymbol{\mu}_i - \boldsymbol{\mu}_j)$. Consequently, the output $\mathbf{m}_{j\to i}$ will also be invariant as the edge operation $\phi_e(.)$ becomes invariant.

Afterward, the equations of our model that update the mean and variance of coordinates $\mathbf{x}$ are $\mathrm{E}(n)$ equivariant as well. In the following, we prove their equivariance by showing that a $\mathrm{E}(n)$ transformation of the input leads to the same transformation of the output. Notice that $\mathbf{m}_{j\to i}$ is already invariant as proven above. Notably, the translation $g$ has no impact over the variance of coordinates $\boldsymbol{\Sigma}_i^{(l)}$. Thus, we want to show:

$$Q\boldsymbol{\mu}_i^{(l+1)} + g = Q\boldsymbol{\mu}_i^{(l)} + g + \frac{1}{|\mathcal{N}(i)|}\sum_{j\in\mathcal{N}(i)}\left(Q\boldsymbol{\mu}_i^{(l)} + g - \left[Q\boldsymbol{\mu}_j^{(l)} + g\right]\right)\phi_\mu(\mathbf{m}_{j\to i}),$$

$$Q^\top \boldsymbol{\Sigma}_i^{(l+1)} Q = Q^\top \boldsymbol{\Sigma}_i^{(l)} Q + \frac{1}{|\mathcal{N}(i)|}\sum_{j\in\mathcal{N}(i)}\left(Q^\top \boldsymbol{\Sigma}_i^{(l)} Q + Q^\top \boldsymbol{\Sigma}_j^{(l)} Q\right)\phi_\sigma(\mathbf{m}_{j\to i}).$$

(9)

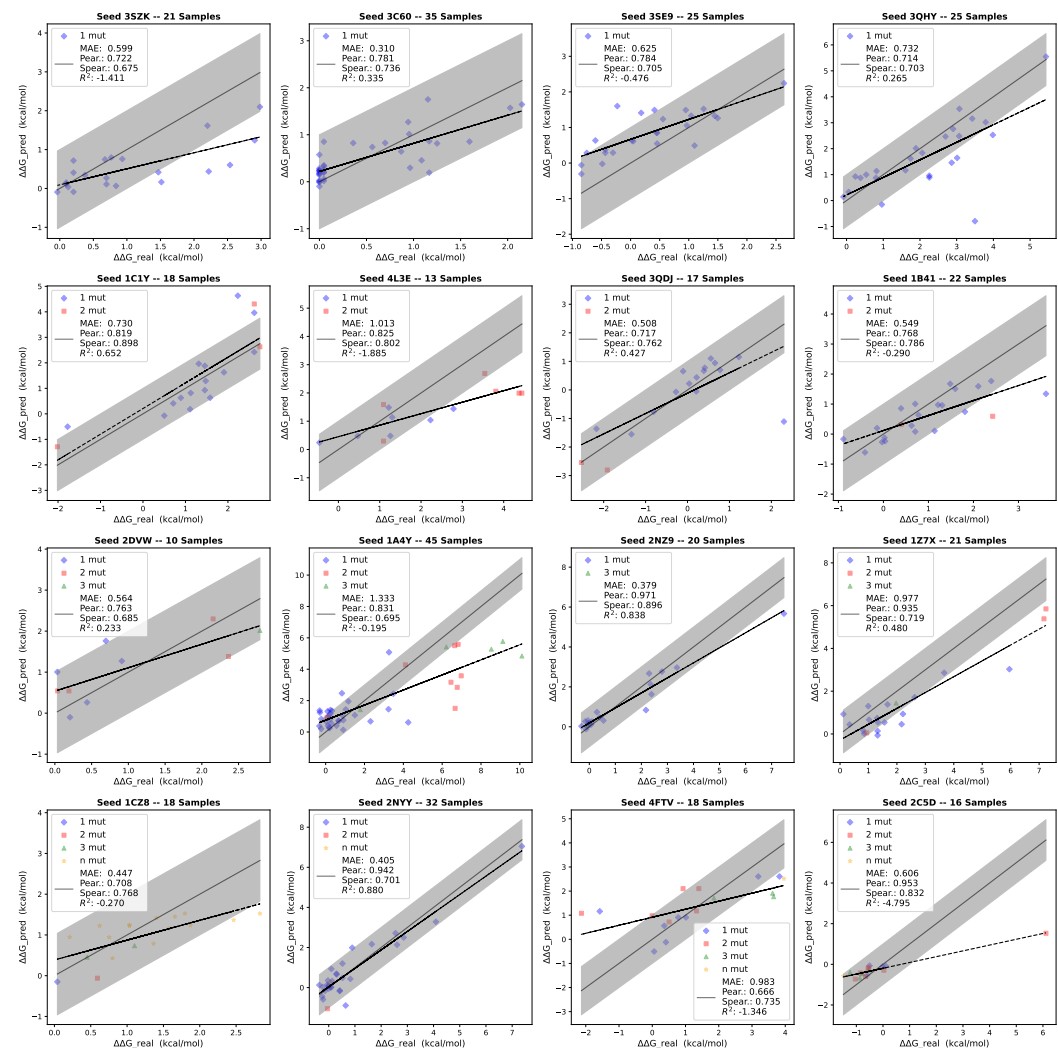

Figure 9: Prediction plots of 16 seed PDBs that are made by Refine-PPI. Four rows correspond to different numbers of mutations, where the gray belt represents acceptable prediction errors. It can be found that Refine-PPI can perform well in all circumstances containing one, two, or more mutations.

Its derivation is as follows.

$$
\begin{aligned}
Q\boldsymbol{\mu}_i^{(l)} + g + \frac{1}{|\mathcal{N}(i)|} &\sum_{j \in \mathcal{N}(i)} \left( Q\boldsymbol{\mu}_i^{(l)} + g - \left[ Q\boldsymbol{\mu}_j^{(l)} + g \right] \right) \phi_\mu(\mathbf{m}_{j \to i}) \\
&= Q\boldsymbol{\mu}_i^{(l)} + g + Q \frac{1}{|\mathcal{N}(i)|} \sum_{j \in \mathcal{N}(i)} \left( \boldsymbol{\mu}_i^{(l)} - \boldsymbol{\mu}_j^{(l)} \right) \phi_\mu(\mathbf{m}_{j \to i}) \\
&= Q \left( \boldsymbol{\mu}_i^{(l)} + \frac{1}{|\mathcal{N}(i)|} \sum_{j \in \mathcal{N}(i)} \left( \boldsymbol{\mu}_i^{(l)} - \boldsymbol{\mu}_j^{(l)} \right) \phi_\mu(\mathbf{m}_{j \to i}) \right) + g \\
&= Q\boldsymbol{\mu}_i^{(l+1)} + g.
\end{aligned}
\tag{10}
$$

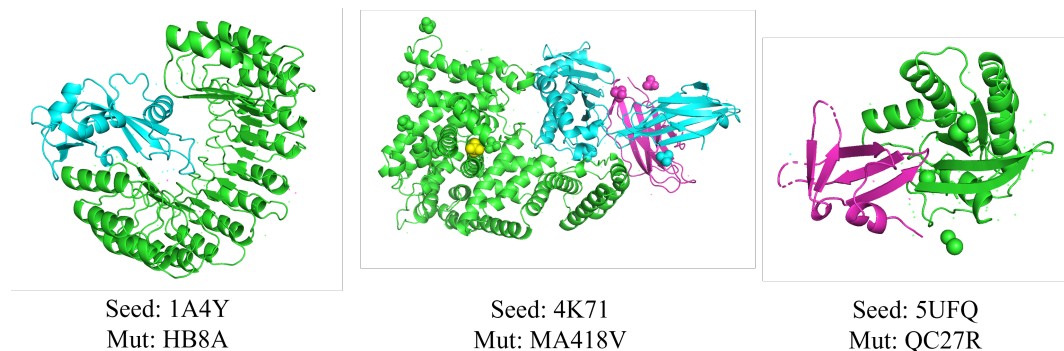

| Seed: 1A4Y | Seed: 4K71 | Seed: 5UFQ |
| Mut: HB8A | Mut: MA418V | Mut: QC27R |

Figure 10: Examples of hallucinated structures of mutation-type.

$$Q^\top \mathbf{\Sigma}_i^{(l)} Q + \frac{1}{|\mathcal{N}(i)|} \sum_{j \in \mathcal{N}(i)} \left( Q^\top \mathbf{\Sigma}_i^{(l)} Q + Q^\top \mathbf{\Sigma}_j^{(l)} Q \right) \phi_\sigma(\mathbf{m}_{j \to i})$$

$$= \mathbf{\Sigma}_i^{(l)} + \frac{1}{|\mathcal{N}(i)|} \sum_{j \in \mathcal{N}(i)} \left( \mathbf{\Sigma}_i^{(l)} + \mathbf{\Sigma}_j^{(l)} \right) \phi_\sigma(\mathbf{m}_{j \to i}) \quad (11)$$

$$= \mathbf{\Sigma}_i^{(l+1)} = Q^\top \mathbf{\Sigma}_i^{(l+1)} Q.$$

Therefore, we have proven that rotating and translating the mean and variance of $\mathbf{x}^{(l)}$ results in the same rotation and translation on the mean and variance of $\mathbf{x}^{(l+1)}$.

Furthermore since the update of $\mathbf{h}^{(l)}$ only depend on $\mathbf{m}_{j \to i}$ and $\mathbf{h}^{(l)}$ which as saw at the beginning of this proof, are $\mathrm{E}(n)$ invariant, therefore, $\mathbf{h}^{(l+1)}$ will be invariant too. Thus, we conclude that a transformation $Q\boldsymbol{\mu}_i^{(l)} + g$ in $\boldsymbol{\mu}_i^{(l)}$ will result in the same transformation on $\boldsymbol{\mu}_i^{(l+1)}$ while $\mathbf{h}^{(l+1)}$ will remain invariant to it so that $\mathbf{h}^{(l+1)}, \left\{ Q\boldsymbol{\mu}_i^{(l+1)} + g, Q^\top \mathbf{\Sigma}_i^{(l+1)} Q \right\}_{i=1}^n =$ PDC-L $\left[ \mathbf{h}^{(l)}, \left\{ Q\boldsymbol{\mu}_i^{(l)} + g, Q^\top \mathbf{\Sigma}_i^{(l)} Q \right\}_{i=1}^n, \mathcal{E} \right]$ is satisfied.

## E  LIMITATIONS AND FUTURE WORK

Despite the success of Refine-PPI in estimating the mutation effect, there is still room left for improvement. First, Refine-PPI keeps most of the complex stable and merely restores a region around the mutant site. It is possible that the entire complex can be significantly different upon mutation. Therefore, a promising future direction would be to enlarge the mask region. Furthermore, previous studies demonstrate the benefit of structural pretraining to dramatically expand the representation space of DL models. We expect to implement MMM with more experimental structures other than PDB (*e.g.*, Alphafold-Database) and transfer the knowledge to predict free energy change.

