# OpenReview forum: "Dynamics-inspired Structure Hallucination for Protein-protein Interaction Modeling"
_ICLR.cc/2025/Conference — ICLR 2025 Conference Withdrawn Submission_

### Official Review · Reviewer_XFf4 · 2024-10-30

**Soundness:** 2
**Presentation:** 2
**Contribution:** 2
**Rating:** 3
**Confidence:** 4

**Summary:**

Summary: The authors propose a method to predict change in binding free energy
of mutated proteins.

**Strengths:**

- The proposed method "PDC-Net" is able to propagate distributions of point clouds through EGNNs, which appears
novel and relevant, including the innate uncertainty values.
I think the paper should revolve around that method and use it for multiple applications.

**Weaknesses:**

- Lack of clarity: It remains unclear what the actual contribution is and what the main theme of the paper is. The title seems to indicate that one main component will be method that produces "halluzations", but later this is hardly picked on. Instead it appears that the masked mutation modeling is a core component, but a bit later again the PDC-Net is crucial. Similarly, the application area remains obscure: is it about predicting protein-protein-interactions or about changes in 3D conformation through mutation or is it about predicting free energies. The paper could be improved a lot by a consistent theme and improved writing.

- Significance: From the application perspective, this work appears very narrow: it can only be applied to antibody-antigen complexes, for which protein mutations are in question. Machine learning methods should be designed to solve a multitude of problems and even application papers should at least aim for broad application areas or point to similar problems in other domains. However, it appears that a very narrow problem is solved by stitching together a couple of machine learning components. The work is also of limited relevance because the performance difference to other methods could just arise by chance.

- Originality: It remains unclear what the novelty of this work is. The method has some novel components, such as the PCD-Net, and a somewhat novel objective. The application, however, is not novel and very narrow. The authors should clearly state what they see as their novelty and then embed it into related works. The authors should state if other methods arise as special cases of their method.

- Technical errors:
a) It appears that Huber-loss breaks the equivariance of the method. Can the authors define Eq (2) clearer because Huber loss is not defined for vectors. Is it applied component wise. Can you theoretically show that equivariance is kept?
b) Results are shown without error bars and confidence intervals (across re-trainings or training sets), Table 1, such that the difference between performance values could just arise by chance. Furthermore non-significant digits are likely shown in Table 1. The authors should perform nested cross-validation (if computationally feasible), or re-run the training multiple times. Ideally the method should also be applied to more than one dataset.

**Questions:**

Included in the "Weaknesses" above.

---

### Official Review · Reviewer_bx9r · 2024-11-01

**Soundness:** 3
**Presentation:** 2
**Contribution:** 3
**Rating:** 5
**Confidence:** 4

**Summary:**

This paper introduces Refine-PPI, a new framework for predicting protein’s mutational effects on protein-protein binding strength. The authors design a masked mutation modeling (MMM) strategy to predict mutant protein structures by training on available wild-type structures, providing crucial geometric information related to the change in binding free energy. They also aim to incorporate the dynamic nature of PPIs into the model by proposing a new kind of geometric GNN called PDC-Net to capture 3D structural dynamics and atomic uncertainty, which is essential for accurately representing the fluctuating nature of PPIs.

**Strengths:**

Compared to previous studies, this paper enables predicting the mutated protein structure and the change in binding free energy simultaneously and also considers the dynamics and flexibility of conformation during the binding process. The study is well-executed and provides detailed experimental results and a comprehensive comparative analysis to demonstrate the model’s improvement over baseline models in predicting free energy changes.  The paper generally provides a clear explanation of the methodology and sufficient background information. The step-by-step breakdown of the Refine-PPI framework is easy to follow, and the authors succeed in describing both the motivation and technical details. The ability to "hallucinate" mutant structures and consider atomic uncertainty could significantly enhance applications in areas where mutant structure data is sparse.

**Weaknesses:**

1. The paper may lack a clear justification for why incorporating dynamic properties into geometric GNNs enhances model performance relative to traditional static approaches. It would be helpful to demonstrate more about the motivations for using PDC-Net and its role in predicting changes in binding affinity.
2. There is a notable formatting issue near Figure 4 and the associated paragraph, where overlapping text obscures readability.
3. The definitions of various notations in Sections 2 and 3 could benefit from greater clarity. Certain variables are not explicitly defined, leading to potential ambiguity. A table summarizing the key notations and their meanings would improve comprehension and ensure that readers can follow the mathematical derivations without confusion.

**Questions:**

1. The paper would benefit from additional validation of the hallucinated mutant structures to demonstrate their contribution to final prediction accuracy. While the concept of generating hypothetical mutant structures is intriguing, the paper lacks sufficient evidence of how these generated structures influence the model’s predictive performance on binding affinity changes.
2. In the ablation study section, the authors introduce two strategies for initializing and corrupting masked regions, but the rationale behind selecting these specific strategies is not sufficiently discussed.
3. I am also wondering if the inclusion of mutation structure in the model will largely increase the computational cost. It would be helpful for the authors to provide some information about their running time comparison.

---

### Official Review · Reviewer_PZNe · 2024-11-02

**Soundness:** 2
**Presentation:** 1
**Contribution:** 3
**Rating:** 1
**Confidence:** 3

**Summary:**

The authors consider protein-protein interactions (PPI) and consider mainly the task of predicting the change in binding free energy (ddG).
They (1) propose a structure refinement module trained by a mask mutation modeling (MMM) task on available wild-type structures and (2) suggest a probability density cloud (PDC) network to take uncertainty, associated with PPI, into account.
Their empirical evaluation, which seems to follow Luo et al. (2023) quite closely seems to show, that their suggested method is superior to many other methods.

**Strengths:**

- relevant research topic
- the suggested probability density cloud (PDC) seems interesting (maybe also for other applications than PPI): I don't remember to have seen such an extension of EGNN (Satorras et al. (2021)) before, but also didn't search extensively, whether it might have been proposed previously already.
- clustering of protein chains in the benchmark (chain cluster split up into train/val/test instead of protein chains themselves)

**Weaknesses:**

The description of the Refine-PPI workflow is unclear at certain points. My best guess is that the authors suggest a denoising task (==MMM) on the wildtype structure for training.
The motivation/post-analysis for MMM did not make it clear for me, why this might be especially useful for ddG prediction.
Why do authors think that learning to denoise wildtype structures is advantagous for better ddG prediction?

Furhter presentation points unclear:
- Figure 3A: It seems that something is masked at the mutant and not at the wildtype. This is possibly in the inference mode. Is there no masking in the inference mode for wildtype, but instead only for mutant type?
- Looking at "Algorithm 1" it is completely unclear what $\mathcal G_t^{\text{WT/MT}}$ or $\tilde{\mathcal{G}}_t^{\text{WT/MT}}$ exactly should be? The link to equation (1) makes it even more confusing as this describes an initialization for **coordinates**, which in Algorithm 1 themselves also seem be denoted as $x_t^{\text{WT/MT}}$.\
==> This is not only about "Algorithm 1" but also all the describing text in the main paper, which describes MMM. Notation is used in a confusing way.\
lines 839-840: Why are coordinate updates computed at all? I don't see them used anywhere later. Only after a long time of guessing how this algorithm might work, it seems that there might be a relationship to $\tilde{\mathcal{G}}^{\text{MT}}$. But this is nowhere made clear, which makes it hard to judge whether the author's contribution might be a meaningful contribution at all.
- One of the most confusing sentences in the paper to me: "It is worth noting that the resulting $x^{\text{WT}}$ does not carry gradients with no backpropagation at this phase". It's completely unclear what the authors want to say with this.
- What exactly is masking in the sense of the authors? Is it equivalent to denoising of coordinates or is the amino acid type somehow also masked by a mask token, which is trained to be predicted? To my opinion denoising should be denoted as denoising and masking should be denoted as masking, but to interchangeably use these terms is confusing. At least I don't see any masking in "Algorithm 1" which is in accordance to usual usage of masking in deep learning.

There seems to be an error in the equivariance proof (equation 11), which might be fixable, but some of the equalities in the equation are questionable. Or is $\Sigma$ always diagonal? ==> then the algorithm notation could be simplified and written without matrices possibly.

It is unclear how PDC integrates with MMM. What is the relationship to Algorithm 1? This is **poorly** explained.

Empirical evaluation:
- There is no reporting of the variance (error bars) in Table 1.
This should be at least be possible for RMSE and MAE across the individual predictions.
Even more interesting would the evaluation according to a cluster-cross-validation procedure to see variablity of correlation and AUROC.
Further statistical tests showing whether Refine-PPI is superior to the other methods would be helfpul to judge whether Refine-PPI is a useful contribution.
- Why is Pearson correlation useful and one of the main metrics? Is there some indication of linear relationship between predictions and ground truth?

Typos:
The paper needs a **massive effort** to improve writing/presentation and formatting. Here 3 other pieces that need to be made better:
- lines 416-417: Figure 5
- chapter A.3 empty: At this place there should be the algorithm.
- Figure 9: What is a seed-PDB? What is an "acceptable prediction error"?

Further Remark:
Luo et al. (2023) was cited in an improper way in the main part of the manuscript. Large parts of Table 1 seem to be copies of their results.

**Questions:**

- Could the authors give a derivation of the expressions in formula (3) or a reference, where exactly these formulas can be found?
- The primary variable is the distance... Is it the squared or the non-squared expression of the norm?

**Details Of Ethics Concerns:**

I was initially thinking of judging the paper with a better score than a strong reject, since I found the PDC component really interesting.

However:
- the carried out benchmark in the main part of the paper looks like the authors put a lot of effort into comparing their method to others, which I saw very positive initially when reading it (i.e., you usually have to think a lot how to compare in a meaningful way with others)

- I think Luo et al. (2023) is especially not properly cited in the main part of the manuscript wrt. Table 1.

I didn't check all the results from Tables 1, 7, 8 in detail but had the impression that many of them are equal to results in the reported tables of Luo et al. (2023), which makes it unplausible for me that the authors obtained these results by own implementations of these methods (which however needs to be **clearly** indicated).
Additionally to their own methods, the authors only seemed to add results from PPIFormer and seem to have a copy error for FoldX in Table 7.

- only in the appendix it is mentioned, that the "descriptions of the implementation" of the baseline methods "follow the same scheme as Luo et al. (2023) and Bushuiev et al. (2023)".

Then the descriptions follow and larger parts of the descriptions seem to be a 1:1 copy of Luo et al, e.g., in the description of ESM-1v, they write exactly as in Luo et al. (2023):
"**We** use the implementation provided in the ESM open-source repository. Protein language models can only predict the effect of mutations for single protein sequences."

To me this was completely irritating, since I initially thought that the authors might in principle have followed a basic "same scheme", but did all the comparison work in principle themselves.
To me it seems that the original authors, who carried out most of the comparisons, are cited in an improper way. If sentences like the above were written like:
"We report the result given by Luo et al. (2023) for which they used the implementation provided in the ESM open-source repository...", then everything would have been clear from the beginning on for me as a reviewer.

To me, this seems like plagiarism. Otherwise to my opinion, it would be fine for ICLR, if papers copy large parts 1:1 from other papers and write one sentence in front of that, which mentions that the "descripton" follows the same "scheme" as the other paper.

---

### Official Review · Reviewer_nyrh · 2024-11-04

**Soundness:** 2
**Presentation:** 2
**Contribution:** 1
**Rating:** 5
**Confidence:** 3

**Summary:**

The authors propose a method of predicting free energy of protein complex formation by using pooled predictions from pre-trained denoising model for wild-type protein and mutated protein. The denoising model is trained on the separate dataset with the goal of recovering the coordinates of the residues beta-carbons of a masked region. The main contribution of this work is introduction of a PDC-Net, which is a generalization of EGNN to the sets gaussian distributions instead of sets of points.
The authors evaluated their approach using the dataset SKEMPIv2 and obtained 0.011 increase (wrt RDE-Net) in Pearson correlation coefficient between ground-truth free energy of association and predictions for different mutations within a single wild-type protein.
They also explored the interpretations of gaussian variance matrix by predicting stable regions of proteins and comparing the variance to the RMSF obtained using MD simulations. They also show that the PDC-Net is better suited at predicting RMSF.

**Strengths:**

The proposed generalization of EGNN to the set of gaussians is a sound idea, backed by the results in section 4.3.3. The method of measuring free energy of association is not exactly novel, but still represents application of ideas from denoising pretraining to the old field of computational structural biology.

**Weaknesses:**

1. Generalization of EGNN to the sets of gaussians is trivial, from the text of the paper it seems that according to the eq. 5, 6 the updates to the average and variance are not coupled, because phi_mu and phi_sigma are two independently learned functions. Appendix B.2 somewhat corrects this approach. However, on lines 250, 262 authors treat functions of gaussian distributions correctly. In general, the introduction of PDC-Net is unclear in whether it treats set of gaussians as probability density distribution or as the abstract mathematical objects transformed with rotations and translations as rigid bodies. In any case clarification is needed and if set of gaussians going into PDC-Net is indeed a probability distribution, it would be nice to have a full derivation of mean and variance updates upon transformation by the EGNN.
2. Proof in appedix D is given only for the case of main text variant of PDC-Net.
3. Lines 238-245 Analogy provided by the authors is rather awkward: atomic position uncertainty is negligible, compared to thermal fluctuations; Quantum numbers just enumerate solutions to the Shrodinger equations for steady state system consisting of electron and an atomic nucleus. I advise removing this analogy completely or asking for help someone more familiar with quantum mechanics/quantum chemistry.
4. The contribution of the PDC-Net itself to the final evaluation result is unclear. What if we substitute PDC-NET for EGNN + standard distances/angles encoding, how will it change the final result?

**Questions:**

Figure 6 caption reads: darker color corresponds to more flexible regions, however it is clear WT 5F4E shows that the core of the upper protein is colored dark(or red). Also this figure should be smoothly shaded without light, otherwise it is exceedingly unclear. Please clarify.

It's unclear what Table 6 shows. Could you please add more precise description of these ablations.

---

### Note · Authors · 2024-11-12

**Comment:**

We sincerely appreciate the time and effort of all reviewers invested in reviewing our manuscript.

**Withdrawal Confirmation:**

I have read and agree with the venue's withdrawal policy on behalf of myself and my co-authors.